# Innate Immune Responses to Influenza Virus Infections in the Upper Respiratory Tract

**DOI:** 10.3390/v13102090

**Published:** 2021-10-17

**Authors:** Edin J. Mifsud, Miku Kuba, Ian G. Barr

**Affiliations:** 1WHO Collaborating Centre for Reference and Research on Influenza, VIDRL, Peter Doherty Institute for Infection and Immunity, Melbourne 3053, Australia; Miku.kuba@influenzacentre.org (M.K.); Ian.Barr@influenzacentre.org (I.G.B.); 2Department of Microbiology and Immunology, University of Melbourne, Melbourne 3053, Australia

**Keywords:** innate immune system, influenza virus, upper respiratory tract

## Abstract

The innate immune system is the host’s first line of immune defence against any invading pathogen. To establish an infection in a human host the influenza virus must replicate in epithelial cells of the upper respiratory tract. However, there are several innate immune mechanisms in place to stop the virus from reaching epithelial cells. In addition to limiting viral replication and dissemination, the innate immune system also activates the adaptive immune system leading to viral clearance, enabling the respiratory system to return to normal homeostasis. However, an overzealous innate immune system or adaptive immune response can be associated with immunopathology and aid secondary bacterial infections of the lower respiratory tract leading to pneumonia. In this review, we discuss the mechanisms utilised by the innate immune system to limit influenza virus replication and the damage caused by influenza viruses on the respiratory tissues and how these very same protective immune responses can cause immunopathology.

## 1. Introduction

Two types of influenza viruses currently co-circulate and cause substantial disease in humans, influenza A and influenza B [1]. Both viruses can be further split, with influenza A viruses having two subtypes A (H3N2) and A (H1N1), and B viruses having two distinct lineages known as B/Yamagata/16/88-like (B/Yam) and B/Victoria/2/87-like (B/Vic) [2]. Infection with any one of these viruses can result in mild respiratory symptoms which are usually confined to the upper respiratory tract (URT) and typically display signs such as fever, sore throat, rhinitis, cough, lethargy, and headache. More severe influenza virus infections are usually associated with lower respiratory tract infections (LRT) leading to viral or bacterial-induced pneumonia, which can be fatal, especially in the elderly [3]. Deaths due to seasonal influenza virus epidemics are estimated to result in 290,000 to 650,000 deaths worldwide [4] depending on the year and which viruses were circulating at that time. Pregnancy and age are two known risk factors, with the very young (<1 year of age), and older adults (>65 years of age) being most vulnerable to severe disease. Other co-morbidities, including asthma, cardiovascular disease, diabetes, and chronic kidney disease, also increase the risk of acquiring severe complications following influenza virus infections [5,6,7].

Sporadic and unpredictable global influenza A virus pandemics have occurred following the introduction of a novel influenza A virus strain that is antigenically distinct from circulating viruses. These viruses are typically introduced from animal reservoirs, where influenza A viruses circulate. The lack of pre-existing immunity to the newly introduced viral strain results in heightened morbidity and mortality in infected individuals, not limited to the aforementioned high-risk groups [8,9].

All influenza viruses are enveloped and contain a negative sense, single-stranded, segmented RNA genome. There are a total of eight RNA segments that encode for RNA polymerase subunits (PB1, PB2, PA), viral glycoproteins hemagglutinin and neuraminidase (HA, NA), viral nucleoprotein (NP), matrix protein (M1), membrane protein (M2) and the non-structural protein (NS1). The HA surface viral glycoprotein, which has a distinct globular ‘head’ and ‘stalk’ structure, facilitates viral entry into a host cell. The NA viral glycoprotein on the other hand facilitates viral release from the host cell. The HA and NA proteins are the most antigenically distinct and are the target of neutralising antibodies elicited via either influenza virus vaccination (antigens) or infection (live virus) [10,11].

### Life Cycle of the Influenza Virus

In humans, the influenza virus replicates in epithelial cells of the upper respiratory tract (URT). Infections of the URT involve the nose, sinuses, pharynx, and larynx. Influenza virus infections generally cause a self-limiting infection in the URT, but the virus does have the ability to replicate in the LRT including the trachea and lungs (Figure 1). To facilitate entry into the host cell, the HA binds to sialic acids present on the epithelial cell surface [12,13]. The virus is then internalised in an endosome, where the acidic environment of the endosome allows the viral M2 ion channel to open [13]. The influx of protons into the virus triggers the fusion of viral and endosomal membranes, facilitating the release of viral RNA into the cytoplasm of the host cell. The viral RNA is transported into the cell nucleus, where transcription occurs. Viral replication occurs through a positive sense intermediate (complementary RNA), which is transcribed into positive sense viral mRNA. This is capped and polyadenylated using host mechanisms, and then exported to the cytoplasm where it is translated [14]. Newly synthesized PB1, PB2, PA (which form the viral RNA-dependent RNA polymerase) and NP protein are transported back to the nucleus to increase the rate of viral RNA synthesis. Viral M1 and nuclear export proteins localise in the nucleus at the later stages of viral infection, bind to viral RNAs and facilitate their export to the cytoplasm. The eight nascent RNA segments acquire ribonucleoproteins (RNPs) and are then bundled together for budding egress [15]. Many of these viral proteins are potential targets for antivirals or vaccines and a number can also modulate the immune response that is generated following infection. In this review, we will discuss these numerous innate immune responses in the URT that attempt to combat the influenza virus and how the virus responds to these actions.

## 2. Innate Immune Responses to the Influenza Virus

To establish an infection and propagate successfully, the influenza virus must evade the innate immune system, whose main role is to prevent or restrict viral replication. In restricting viral replication to the URT and not allowing entry into the LRT, vital respiratory functions such as gaseous exchange are not impaired. In addition, the innate immune system plays an important role in activating the adaptive immune response which is crucial in clearing any remaining influenza virus or infected cells, resolving the infection and generating memory cells that can help the host better respond to future influenza virus infections. For example, cytokines released from infected cells recruit dendritic cells (DCs) into the URT. DCs are required for activation of influenza-specific adaptive immune responses that facilitate viral clearance and the resolution of inflammation. The importance of the adaptive immune response during influenza virus infections are thoroughly reviewed elsewhere [4,8,16] and will only be briefly mentioned in this review [17,18,19].

### 2.1. Trapping the Influenza Virus in the Mucous Layer

The respiratory epithelium is coated with a mucous layer, which is a multicomponent viscous or a soft elastic fluid [20]. The more viscous mucus gel layer sits on top of the periciliary liquid layer (PCL), where the cilia lie. The PCL height extends to the same level as the cilia. To facilitate the beating of the cilia, the lower layer of the PCL is less viscous than the top layer. The importance of the mucous layer during influenza virus infections was first noted in 1940s and 1950s [21,22]. The influenza virus, much like other pathogens, becomes trapped non-specifically in the mucus layer and is expelled via mucocillary clearance before it is able to infect underlying cells [23].

#### Mucins and Their Inhibitory Function against the Influenza Virus

Mucins make up a significant proportion of the mucus layer and act as a barrier to the influenza virus infecting underlying epithelial cells [24]. Mucins are a family of high molecular weight, heavily glycosylated proteins that contain complex glycan chains, mainly consisting of O-glycans to which N-acetylgalactosamine (GalNAc) is added, followed by the additional glycan moieties galactose, N-acetylglucosamine (GlcNAc), fucose, sialic acids α2,6 for mammalian cells and others [25]. The influenza virus becomes trapped in the mucous layer by binding sialic acid on the surface of mucins [26,27]. To overcome the mucin block and infect the underlying cells, the viral NA cleaves sialic acids, which prevents the trapping of the virus in the mucous [28,29,30]. In vitro studies have shown that MDCK cell cultures overlayed with human mucous were able to inhibit both swine and human influenza A viruses [25,31,32]. The importance of NA in avoiding the mucus layer was also shown using viruses with low NA activity which became trapped in the mucus layer in ferrets and failed to transmit to other susceptible ferrets [27,33].

### 2.2. The Role of Soluble Proteins against Influenza Virus Infection

Soluble innate immune inhibitors play an important role in blunting influenza viral replication and modulating the immune response to viral infection. In the nasopharynx, these inhibitors include surfactant proteins (SP)- and D and α/β-defensins that will be discussed in this section.

#### 2.2.1. The Antiviral Properties of Surfactant Protein A and D

SP-A and SP-D form trimeric units consisting of four domains, a C-terminal carbohydrate binding domain (CRD), an α-helical coiled-coil neck, a collagenous domain, and an N-terminal domain [34]. These proteins are known as collectins because they contain collagen and are functional lectins, which bind carbohydrates in a calcium dependent manner using their CRD. The functional differences between these two molecules is described elsewhere [35].

SP-D neutralises the influenza virus through interaction with high mannose oligosaccharides located near the HA binding site [36]. Conversely, SP-A occupies the HA binding site through its own salicylic acid, found on the asparagine 187 residue of the CRD [37]. Using different mechanisms to inhibit the HA, both SP-D and SP-A prevent the influenza virus from infecting epithelial cells. Studies in mice have shown that SP-A and SP-D knock-out mice are more susceptible to influenza virus infection, elevated viral load and increased infiltration of inflammatory cells and cytokines when compared to wildtype mice [38,39,40]. The administration of exogenous SP-A or D restored the protective effects of these proteins [38,39,40].

Inhibition of the influenza virus by these surfactant proteins is dependent on the presence of glycans on the viral HA. Different influenza A virus strains such as those which arise via zoonotic events have very little glycosylation on the HA and as such, hence they can avoid recognition by these proteins. For example, in mouse studies, the 1918 and 2009 pandemic H1N1 and highly pathogenic avian H5N1 influenza virus strains are resistant to inhibition by these proteins due to the lack of glycosylation on their HA [41,42].

#### 2.2.2. The Role of α/β-Defensins

Defensins are a major class of antimicrobial peptides found in pulmonary secretions that have strong neutralising activity against a range of influenza viral strains [43,44]. One group of α-defensins is packaged in neutrophil granules and these are termed human neutrophil peptides. In contrast to the surfactant proteins, α-defensins do not directly inhibit HA activity. They do, however, interact with the virus and induce viral aggregation, inhibiting their infectivity [43,44,45] and α-defensins also promote neutrophil uptake of viral aggregates resulting in the destruction of the virus [45]. Others have also shown that α-defensins inhibit the influenza virus through binding to epithelial cells and inhibiting protein kinase C (PKC) activation [46], which is essential in RNP assembly during primary genome replication [47].

Respiratory epithelial cells secrete β-defensins constitutively and in response to influenza virus infection [48]. Despite being less potent inhibitors of the influenza virus when compared to α-defensins, β-defensins have important immunomodulatory roles during influenza virus infection [49]. Mouse β-defensin is a homolog of human β-defensin [50] and, therefore, the mouse model is used to investigate their role during viral infection. Genetically modified mice lacking β-defensins infected with a mouse-adapted strain of the influenza virus (A/H1N1/PR/08/1934, A/Puerto Rico/8/34 (PR8)) had viral burdens comparable to wild-type animals [51]. In spite of comparable viral loads, animals lacking β-defensins exhibited severe weight loss and greater inflammation in the lungs and had reduced survival rates when compared to their wild-type counterparts [51]. This study demonstrates the potent ability of these defensins to modulate the immune response to the influenza virus without directly inhibiting virus replication.

### 2.3. Pattern Recognition Receptors (PRRs) and Influenza Virus

There are three major PRRs, which are arguably the most important activators of the innate immune system, responsible for sensing the influenza virus. The influenza virus is recognised by at least three different classes of PRRs. The triggering of these PRRs by influenza virus and subsequent activation leads to a range of cell-intrinsic antiviral signals as well as the production of cytokines and chemokines.

#### Recognition of the Influenza Virus via Toll-like Receptors

Toll-like receptors (TLR), of which there are 10 different types in humans, are receptors that can recognise a wide range of bacterial, fungal and viral products (Figure 2). There are three different TLRs responsible for the recognition of the influenza virus: TLR-3, 7 and 8.

TLR-3, like TLR-7/8, is in the endosome. However, unlike the latter two, it is also located at the cell surface of bronchial and alveolar epithelial cells. In addition, unlike other TLRs [52], TLR-3 only utilises a toll-interlukin receptor-domain-containing adapter protein-inducing interferon (IFN)-b (TRIF) [52], whereas other TLRs activate myeloid differentiation primary response-88 (MYD88). Activation of TLR-3 and subsequently TRIF ultimately leads to the production of pro-inflammatory cytokines and type 1 IFNs. The signaling pathways triggered by influenza virus are complex and beyond the scope of this review but are detailed elsewhere [53,54].

TLR-3 recognises double-stranded RNA, and the influenza virus genome is single stranded [55], with any double-stranded RNA replication intermediates rapidly degraded by RNA helicases [56]. Therefore, the exact structures of the influenza virus recognised by TLR-3 are not known [55]. The importance of TLR-3 is evident in mice lacking this receptor, which have a nine times higher viral burden when compared to wild-type mice infected with A/H3N2 mouse-adapted virus [57]. In humans, a mis-sense mutation F303S of the TLR-3 gene is linked with influenza-associated encephalopathy, a neurological consequence of severe influenza virus infection [58]. Upon further investigation in vitro, TLR-3 receptors encoding this mutation were shown to have impaired ability to activate the transcription factor NFκB [58], suggesting that this polymorphism could facilitate extra-pulmonary spread of the influenza virus by failing to activate TLR-3 and its downstream signaling pathways. Another important polymorphism in the TLR 3 gene associated with increased disease severity is rs5743313 (genotype C/T). This single nucleotide polymorphism (SNP) was identified in children with severe influenza pneumonia [59]. This SNP was less prominent in children with mild disease [59], where other mechanisms are able to compensate.

Similarly to how TLR-3, TLR-7/8 are localised inside the endosome of most cells and recognise influenza viral RNA, [60] TLR-7 is responsible for the recognition of single-stranded RNA, where, due to the endosomal location of this receptor in respiratory epithelial cells, the influenza virus is recognised during viral entry. Once activated, TLR-7 and TLR-8 induce MYD88, resulting in the activation of NFκB and interferon regulatory factor (IRF) depending on the cell type. For example, activation of bronchial epithelial cells results in the production of interleukin-6 (IL-6) and Type III IFNs [61], whereas activation of TLR-7 or TLR-8 in plasmacytoid DCs produces high levels of type I IFNs via IRF7 [62] (Figure 2).

### 2.4. Recognition of the Influenza Virus via RIG-I Like Receptors

Influenza viral RNA can also be detected in the cytoplasm by RIG-I-like receptors (RLR), a family of three RNA sensing proteins: LGP2, MDA5 and RIG-I, which are expressed in epithelial cells, macrophages and conventional DCs [63]. Upon sensing the 5′ triphosphate RNA promoter region of the intact genomic segment or shorter fragments, the expression of RIG-I is enhanced [55,64]. Moreover, RIG-I preferentially associates with shorter viral RNA molecules or sub-genomic defective interfering particles [65]. Following recognition of viral RNAs, the RIG-I helicase domain binds to ATP, and forms a complex with the mitochondrial antiviral signaling protein (MAVS) via caspase recruitment domains. MAVS signaling leads to the production of Type I IFNs and pro-inflammatory cytokines through IRF and NFκB, respectively (Figure 2). Ducks ubiquitously express RIG-I in mucosal tissues, which almost certainly contributes to their immunity against influenza virus disease. Ducks are a natural reservoir of H5N1 and H7N9, where these viruses reside in their gastrointestinal tract causing asymptomatic infections. However, in chickens that lack the RIG-I receptor infections with these viruses are highly pathogenic and usually fatal [66,67,68]. During the 2009 H1N1 human pandemic, reduced antiviral responses and severe disease was identified in a patient with SNPs in the CARD and RNA binding domain of RIG-I [69]. To compensate for the reduced antiviral response, the pro-inflammatory cytokine response was elevated, indicative of a dysregulated immune response and heightened immunopathology [69].

### 2.5. Pro-Inflammatory Cytokines and Inflammasomes

Activation of NFκB due to recognition by TLRs leads to the production of pro-inflammatory cytokines and chemokines such as IL-1β, tumor necrosis factor (TNF-α), IL-6, IL-12 and IL-8 [70], which are essential in the recruitment and activation of innate immune cells, such as DC’s, neutrophils, monocytes, macrophages. The production of TNF-α and IL-1β are involved in a positive feedback loop, which leads to further activation of NFκB and cytokine production [71]. During infection with seasonal influenza viruses, these cytokine and chemokine responses are sufficient to facilitate viral clearance. However, infection with H5N1 influenza viruses is characterised by hyperinflammatory cytokine and chemokine responses, which are associated with increased mortality in mice and macaques [72,73]. The exaggerated cytokine response induced during H5N1 influenza virus infections causes pulmonary congestion and ultimately, compromises airway gas exchange leading to death [17]. To further elucidate the role of cytokines and chemokines in H5N1 infections, genetically modified mice that were unable to produce a specific cytokine IL-6 or MCP-1 or TNF-α were infected with H5N1 virus [74]. These mice were not protected from mortality associated with infection, suggesting that the immunopathology caused following viral infection is a multi-factorial process, which cannot solely be due to or reversed by the elimination of just one inflammatory cytokine.

The exact mechanisms regulating the cytokine responses following influenza virus infection are not fully understood. However, there is increasing evidence to suggest that inflammatory disease may be influenced by inflammasome activation [75,76]. Inflammasomes are multimeric protein complexes that assemble in the cytosol after sensing the influenza virus via Nod-like receptor protein-3 (NLRP3) [77]. Inflammasome activation requires two signals: the first is an initial priming step during which the virus is recognized by TLR-3, TLR-7, and/or RIG-I [78]. Activation of signal 1 in the inflammasome pathway produces pro-forms of IL-1β, IL-18 and Caspase-1. This leads to other components of the inflammasome pathway complex being upregulated, such as apoptosis-associated speck-like protein containing CARD (ASC) and signal 2 inflammasome receptors NLRP1, NLRP2, NLRP4, NLRC4 or AIM2 [79]. Activation of NLRP3, providing the second signal, is achieved by either recognition of viral RNA, M2 protein on the virus membrane or PB1-F2 (a PB1 gene segment) [76,79,80,81]. This second and final signal required for inflammasome activation leads to the oligomerization and auto-proteolytic cleavage of caspase-1. Active caspase-1 then cleaves the pro-forms of IL-1β and IL-18, which are then secreted from the cells, causing immune cells such as neutrophils to be recruited into the area [82]. Additionally, inflammasome activation causes the rapid, pro-inflammatory form of cell death known as pyroptosis [83], further limiting viral propagation, and dissemination (Figure 2).

In mice, activation of the NLRP3-inflammasome during infection with PR8 was essential in limiting lung damage [79,84]. Mice deficient in inflammasome proteins were shown to have reduced survival rates associated with fatal pneumonia, due to a more permeable epithelial cell layer facilitating lung oedema and alveolar fibrosis [84]. This illustrates the critical balance required for the controlled production of inflammatory cytokines IL-1β and IL-18, as exacerbated or prolonged stimulation adds to the disease burden, while appropriate activation is required to facilitate viral clearance.

### 2.6. Interferon Stimulated Genes and Their Role in Influenza Virus Suppression

Type I and III interferon (IFN) pathways rapidly promote an antiviral state by inducing the expression of hundreds of genes, grouped as IFN-stimulated genes (ISGs) [85]. These genes limit viral replication and spread, but most importantly induce immune responses in neighboring cells, which in turn protects them from influenza virus infection. Type 1 and type III IFNs mediate different roles during influenza viral infections [86,87]. For example, type I and III IFN receptor-deficient mice were unable to control viral burden, but only type III IFN receptor-deficient animals rapidly transmitted the virus to naïve contact mice, when compared to wild-type animals or mice lacking functional type I IFN receptors [87]. The potential deleterious effects of type 1 IFNs were demonstrated in the dual combination knockout mice. Recombination activating gene-1 (RAG-1) mice have no mature T or B-cells, infections of these mice with influenza A virus resulted in death 14 days after infection [86]. RAG 129 mice lack mature T and B-cells as well as IFN α/β receptor, following infection, these animals succumbed to infection earlier than RAG-1 mice. Viral loads in the lungs of mice were comparable between animals [86]. Heightened mortality was attributed by the authors to the elevated levels of type I and III IFNs.

During downstream IFN production, ISG-derived proteins perform two main functions. Firstly, they directly limit viral replication by shutting down protein synthesis [88], and triggering apoptosis [89]. Secondly, ISGs activate key components of the innate and adaptive immune systems, including antigen presentation and production of cytokines involved in T and B-cell activation (discussed in [88]). Several hundred ISGs were identified and three key genes, myxovirus resistance protein (Mx), IFN-induced transmembrane protein 3 (IFTM3), and IRF7 are important in the role of during influenza virus infections.

Mx proteins are a family of GTPases with broad antiviral activity against influenza viruses [90,91] that inhibit viral transcription and replication by inhibiting viral nucleocapsid entry into the nucleus [92,93,94]. Conserved transmembrane protein, IFITM3, restricts the proliferation of the influenza virus by modulating endosomal cholesterol. This protein blocks pH-dependent fusion during late endosomal entry of the virus into the cytoplasm [95,96,97]. IFITM3 is constitutively expressed by respiratory epithelial cells and is postulated to be a mechanism by which the respiratory epithelium may reduce viral dissemination [98]. Mice deficient of IFITM3 had higher viral loads and experienced greater morbidity and mortality when compared to wildtype mice infected with the influenza virus [96,97]. SNPs in human IFTIM3 (rs 12252) associated with severe disease following influenza virus infection, indicating the importance of this ISG [99].

IRF7 is a master regulator of type I IFNs, with levels in respiratory epithelial cells usually being low but they increase in response to IFN signaling [100]. Respiratory epithelial cells obtained from a child with an IRF7 null mutation failed to generate an IFN response and this was linked to the death of the child [101]. The important role of the positive feedback loop generated by IFN and IRF7 was also observed in mice, as influenza-infected IRF7 knockout mice had greater morbidity and mortality and significantly reduced IFN response when compared to wild-type mice [102].

Transcriptomic analysis of the lungs and ileum samples of H5N1 infected ducks and chickens further elucidated the importance of RIG-I and ISG. Ducks that express RIG-I rapidly increased ISG expression following infection, whereas chickens lacked receptor upregulated expression of genes that were responsible for T and B-cell activation [103,104]

### 2.7. The Role of Macrophages, Monocytes and Dendritic Cells during Influenza Virus Infection

Macrophages play an important role in the outcome of influenza virus infections. Macrophages are abundant in the LRT but are less likely to be involved in uncomplicated cases of influenza virus infection where the virus typically remains in the URT [105]. Their role in the LRT following influenza virus infection is significant where they play a pivotal role in eliminating the virus and trigger wound repair following viral infection. This is reviewed elsewhere [106].

Conversely, monocytes are mucosal sentinels and rapidly infiltrate the URT following influenza virus infection [107]. Monocytes are relatively resistant to influenza-induced cell death,, [108] but when infected with influenza virus, results in the production of chemokines and cytokines such as MCP-1, IL-6 and IL-8 [107]. In vitro studies have also shown that influenza virus infection facilitates the differentiation of monocytes into monocyte-derived dendritic cells [109]. Despite, their importance in recognising the influenza virus and ‘sounding the alarm’ increased numbers of monocytes in the URT are used to predict disease severity in patients following influenza virus infection [110].

Dendritic cells (DCs) are professional antigen presenting cells which bridge the gap between the innate and adaptive immune systems [111]. DCs can be differentiated into many different cell types but the most important in the URT are CD103+ DCs. CD103+ DCs are efficient antigen presenting cells that are constantly surveying the URT, once activated these cells migrate to the draining lymph node [112]. In the lymph node these cells efficiently cross present antigens to CD8+ T cells [113,114]. Depletion of CD103+ cDCs, in mice led to exacerbated disease severity when compared to animals with intact CD103+ cDCs [115], suggesting these cells play a crucial role in influenza virus infection. The role of DCs are further reviewed in [111].

## 3. Conclusions

Seasonal influenza A and B viruses are a major health burden for humans and there is an ongoing threat of new pandemic influenza viruses emerging. The first barrier to influenza virus infections is the innate immune system which is especially important for the very young who have not encountered the influenza virus previously and have not formed an effective adaptive immune response that can prevent or ameliorate influenza virus infections. The innate immune system is a complex system of cellular and extra-cellular components limit viral replication and dissemination to the LRT. The innate immune system is an evolutionary defence mechanism present in most multicellular organisms and as a result there are extensive redundancies throughout species. The studies described in this review have shown that the absence of one or more components of the innate immune system can have deleterious effects on the antiviral response. In contrast, the overzealous activation of the innate immune system can cause immunopathology that may result in poorer outcomes and can be fatal, demonstrating the fine balance between protection and immunopathology. Immunomodulating agents targeting the innate immune system have been studied extensively as potential prophylactic treatment options but getting the balance right between activating an innate immune pathway sufficiently but not over stimulating it so that immunopathology is the outcome is challenging.

## Figures and Tables

**Figure 1 viruses-13-02090-f001:**
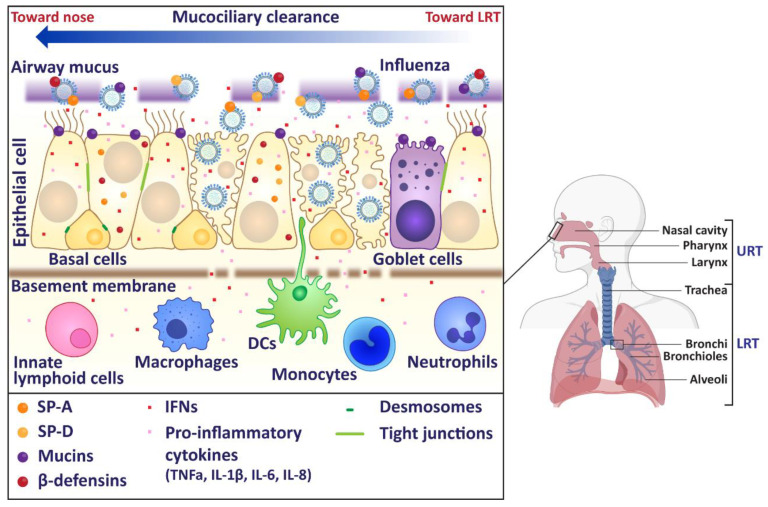
Schematic of human upper respiratory tract. In humans, the upper respiratory tract (URT) consists of the nose, sinuses, pharynx, and larynx. The lower respiratory tract (LRT) includes as the trachea and lungs. Influenza virus infection primarily occurs in the URT, where the virus must first overcome the mucus layer to infect underlying epithelial cells. Influenza virus infections generally cause a self-limiting infection in the URT, but the virus does have the ability to replicate in the trachea and the lungs. Image made using BioRender.

**Figure 2 viruses-13-02090-f002:**
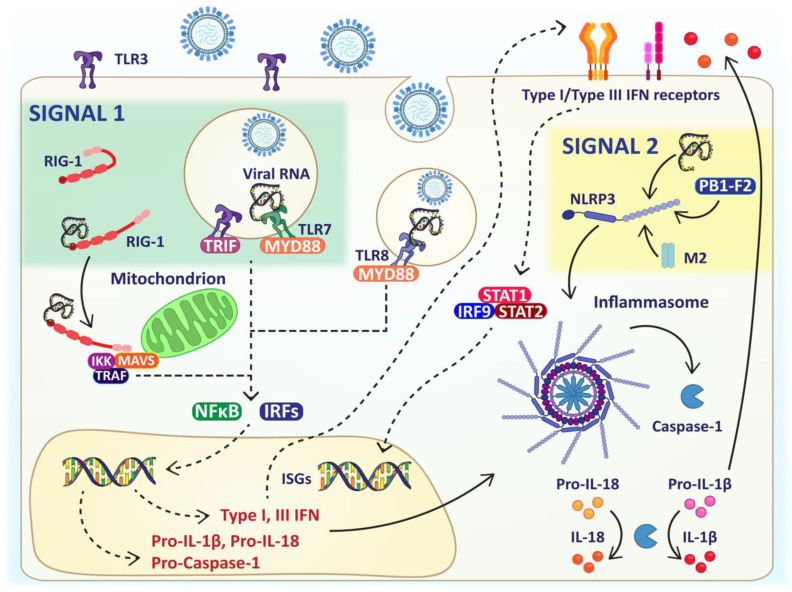
Localisation of PRRs responsible for influenza viral recognition. Schematic of PRRs responsible for viral recognition. The inflammasome requires two signals for activation and are shaded green and yellow, respectively. Image made using BioRender.

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
