# Peer review of "Innate Immune Responses to Influenza Virus Infections in the Upper Respiratory Tract"

_viruses, 2021, doi:10.3390/v13102090_

Round 1

Reviewer 1 Report

Innate immune responses to influenza viral infections in the upper respiratory tract  Edin J Mifsud1,2* Miku Kuba1, and Ian Barr1,2

The authors should clarify what their review bring to science literature that has not been published recently

Front. Cell. Infect. Microbiol. | https://doi.org/10.3389/fcimb.2020.563850

Biomedical Journal, Volume 42, Issue 1, 2019, Pages 8-18, ISSN 2319-4170, https://doi.org/10.1016/j.bj.2018.12.009

Viruses 2020, 12(7), 755; https://doi.org/10.3390/v12070755

The terminology around influenza and influenza viruses need to be substantially revised. 

Author Response

The authors should clarify what their review bring to science literature that has not been published recently

Front. Cell. Infect. Microbiol. | https://doi.org/10.3389/fcimb.2020.563850

Biomedical Journal, Volume 42, Issue 1, 2019, Pages 8-18, ISSN 2319-4170, https://doi.org/10.1016/j.bj.2018.12.009

Viruses 2020, 12(7), 755; https://doi.org/10.3390/v12070755

We thank the reviewer for their request for clarification. This review focuses on key events that occur in the upper respiratory tract during influenza infection. The reviews listed above discuss key events primarily involved in lung, or lower respiratory tract responses. Additionally, we discuss the role of single nucleotide polymorphisms in humans that have been shown to influence disease outcomes, which are also not discussed in the aforementioned reviews.

Other points for review:

Influenza A/B/C/D throughout, clarify that the review is about influenza (disease) caused by influenza A viruses in humans with reference to animal disease / infections by ‘flu A viruses as required.

Influenza is the disease, Influenza virus causes infection (agent). The text should be refined to ensure correct use / terminology.

We thank the reviewer for this comment and have modified the main text accordingly to clarify this issue.

Ln 77 Eight viral ribonucleoproteins, the nascent RNA segments (eight) acquire the RNPs and are then bundled together for budding egress

This line has been changed and now reads “The eight nascent viral RNA segments acquire ribonucleoproteins (RNPs) and are then bundled together for budding egress”

Ln 112-113 Cell receptor SA – alpha 2,6 mammalian and others

This line has now been changed and reads “……. sialic acids (via α2,6 linkages for mammalian cells) and others”

First and subsequent use of abbreviations e.g LRT, e.g. Ln 61, 87

Lower respiratory tract has been removed and has been replaced with LRT

Ln 43/44 influenza viruses from animal to humans (Flu A yes and selected subtypes, others rare)

“A” has been added to the sentence and it reads “Sporadic and unpredictable global influenza A virus pandemics have occurred following the introduction of a novel influenza A virus strain that is antigenically distinct from the circulating viruses”

Ln 88 incomplete sentence

Sentence has been completed and now reads “ Dendritic cells are ….”

 Ln 145 Caution when switching from interpretation of animal evidence to humans

“In mouse studies” has been added to the sentence to clarify

 ln 116-119 Use of TIR and TRIF etc abbreviations without full version (non-specialist readers), check all

TIR- toll-interleukin receptor and TRIF are defined in the text see line 190

Ln 192 triggered by influenza virus, not disease

Virus has been added to the sentence

Ln 232, HPAIV in ducks – incorrect see Alexander, Vet Microbiol 2000 May 22;74(1-2):3-13. doi: 10.1016/s0378-1135(00)00160-7. Duck LP >> chicken HP

We thank the reviewer for their comment and we have clarified the statement

Ln 235, 2009 H1N1 pandemic statement applies to humans, not swinepdmH1N1 – host clarity in sentences.

Human has been included in the statement

Ln249 HPAI abbreviation already used

The definition of the abbreviation has been removed

Ln 249-255 H5N1 infection of humans, mice or birds, clarify?

In mice and macaques has been added to the end of the sentence

Reviewer 2 Report

In this review by Mifsud, Kuba and Barr, authors review the role of innate immune responses to influenza virus from the context of innate resistance and immunopathology. This is a succinctly-written review (not a thorough and detailed) that is topical and provides up-to-date information on the state of the field. Authors provide just enough details about various facets of innate immunity including mucus, respiratory epithelial cells, viral sensing, surfactant proteins and Type I IFNs.

  1. The authors should consider discussing studies influenza viral control and morbidity in RAG-deficient and athymic mice.
  2. It would be appropriate to discuss the role of macrophage and DC subsets (inflammatory macrophages, alternatively activated macrophages, TIP DCs etc.) in driving pulmonary inflammation and wound healing responses.
  3. Authors need to discuss recent work from Paul Thomas’s group about the different role of cell subtypes in lungs (including fibroblasts, stromal cells etc.) in regulating the inflammatory milieu in influenza virus-infected lungs.
  4. Figure 1 show neutrophils and monocytes and should include macrophages, innate lymphoid cells etc.
  5. In line 68, it is not accurate to suggest that translation occurs in nucleus.
  6. Sentence in Line 88 needs editing.
  7. Text in lines 99 and 100 needs to be rephrased to improve clarity.

Author Response

  1. Ln 292 mammalian flu A infections The authors should consider discussing studies influenza viral control and morbidity in RAG-deficient and athymic mice.

We thank the reviewer for their comments and we have added a section discussing RAG-deficient mice.

Starting on line 292

‘The potential deleterious effects of Type 1 IFNs was demonstrated in the dual combination knockout mice. Recombination activating gene-1 (RAG-1) mice have no mature T or B-cells, infections of these mice with influenza A virus resulted in death 14 days after infection {Davidson, 2014 #107}. RAG 129 mice lack mature T and B-cells as well as IFN α/β receptor, following infection these animals succumbed to infection earlier than RAG-1 mice. Viral loads in the lungs of mice were comparable between animals {Davidson, 2014 #107}. Heightened mortality was attributed by the authors to the elevated levels of Type I and III IFNs.’

We have not included a statement in regards to athymic mice as these animals lack CD8 T-cells and the adaptive immune response was beyond the scope of this review.

  1. It would be appropriate to discuss the role of macrophage and DC subsets (inflammatory macrophages, alternatively activated macrophages, TIP DCs etc.) in driving pulmonary inflammation and wound healing responses.

We have added a section into the review discussing immune cells and their role during influenza infections.

2.7 The role of macrophages, monocytes and dendritic cells during influenza virus infection

Macrophages play an important role in the outcome of influenza infections. Macrophages are abundant in the LRT but are less likely to be involved in uncomplicated cases of influenza virus infection where the virus typically remains in the URT {Vangeti, 2018 #2}. Their role in the LRT following influenza infection is significant where they play a pivotal role in eliminating the virus and trigger wound repair following viral infection.  This has been reviewed elsewhere {Nicol, 2014 #1}.

Conversely, monocytes are mucosal sentinels and rapidly infiltrate the URT following influenza virus infection {Gill, 2008 #3}. Monocytes are relatively resistant to influenza-induced cell death {Hoeve, 2012 #216} but when infected with influenza virus results in the production of chemokines and cytokines such as MCP-1, IL-6 and IL-8 {Gill, 2008 #217}. In vitro studies have also shown that influenza infection facilitates the differentiation of monocytes into monocyte-derived dendritic cells {Hou, 2012 #215}. Despite, their importance in recognising the influenza virus and ‘sounding the alarm’ increased numbers of monocytes in the URT have been used to predict disease severity in patients following influenza virus infection {Oshansky, 2014 #214}.

Dendritic cells (DCs) are professional antigen presenting cells which bridge the gap between the innate and adaptive immune systems {Waithman, 2012 #92}. DCs can be differentiated into many different cell types but the most important in the URT are CD103+ DCs. CD103+ DCs are efficient antigen presenting cells that are constantly surveying the URT, once activated these cells migrate to the draining lymph node {Ho, 2011 #96}. In the lymph node these cells efficiently cross present antigens to CD8+ T cells {Helft, 2012#97;del Rio, 2007 #98}. Depletion of CD103+ cDCs, in mice led to exacerbated disease severity when compared to animals with intact CD103+ cDCs {GeurtsvanKessel , 2008 #95}, suggesting these cells play a crucial role in influenza virus infection. The role of DCs have been further reviewed in {Waithman, 2012 #92}.

  1. Authors need to discuss recent work from Paul Thomas’s group about the different role of cell subtypes in lungs (including fibroblasts, stromal cells etc.) in regulating the inflammatory milieu in influenza virus-infected lungs.

We again thank the reviewer for their comments. We have specifically not included any information in regards to the responses that occur in the lungs/lower respiratory tract (which Paul Thomas’ article does) as they were beyond the scope of this review, which focuses on the innate immune responses to influenza in the upper respiratory tract.

  1. Figure 1 show neutrophils and monocytes and should include macrophages, innate lymphoid cells etc.

Macrophages and innate lymphoid cells have been included in figure 1

  1. In line 68, it is not accurate to suggest that translation occurs in nucleus.

We have changed this line to read “The viral RNA is transported into the cell nucleus, where transcription occurs”

  1. Sentence in Line 88 needs editing.

The sentence has been changed and now reads “In restricting viral replication to the URT and not allowing entry into the LRT, vital respiratory functions such as gaseous exchange are not impaired.  In addition, the innate immune system plays an important role in activating the adaptive immune response which is crucial in clearing any remaining influenza virus or infected cells, resolving the infection and generating memory cells that can help the host better respond to future influenza infections.”

  1. Text in lines 99 and 100 needs to be rephrased to improve clarity.

This line has been changed and now reads “‘The PCL height extends to the same level as the cilia. To facilitate the beating of the cilia, the lower layer of the PCL is less viscous than the top layer.’

Round 2

Reviewer 1 Report

The manuscript has been considerably improved and provides a useful review of innate immune response to influenza virus infection, with or without development of the disease in humans, or animal models of human influenza disease. 

the title should include ‘influenza virus infection’

ditto lines 55, 88, 230, 336, 339, 346

Line 55 influenza vaccine (antigen, virus etc) or infection (live virus)

the introduction mentions influenza A and influenza B viruses, this should be consistent throughout the text, influenza A viruses infect many animals whereas Influenza B viruses circulate widely only among humans. Care should be taken through out the text to ensure it is clear which ‘flu viruses the authors are referring to. 
Lines - 43, 115-6, etc

Line 29, symptoms / clinical signs differentiation 

Line 144 Influenza A viruses are zoonotic influenza B are not 

Line 231 but space in 

Line 233 change huamn to human 

Line 247 highly pathogenic avian influenza is a state only in birds (definition), HOAIVs can infect humans, zoonotic, but the HP status is not relevant in humans 

Line 316 Ifitm3 should be capital letters as per other instances, there is also a b instead of beta in the text, editorial checks 

conclusions section, influenza A/B or both refine 

Author Response

the manuscript has been considerably improved and provides a useful review of innate immune response to influenza virus infection, with or without development of the disease in humans, or animal models of human influenza disease. 

We thank the reviewer for their comments and for also going through the manuscript again our responses are below in blue. And have also been made in the text in blue.

the title should include ‘influenza virus infection’

Virus has been added to the title

ditto lines 55, 88, 230, 336, 339, 346

Virus has been added to these lines also

Line 55 influenza vaccine (antigen, virus etc) or infection (live virus)

These have been added to the text

the introduction mentions influenza A and influenza B viruses, this should be consistent throughout the text, influenza A viruses infect many animals whereas Influenza B viruses circulate widely only among humans. Care should be taken through out the text to ensure it is clear which ‘flu viruses the authors are referring to. 
Lines - 43, 115-6, etc

We have added influenza A viruses to this section

Line 29, symptoms / clinical signs differentiation 

We have change this to signs

Line 144 Influenza A viruses are zoonotic influenza B are not 

Have added influenza A to this section

Line 231 but space in 

Space has been added

Line 233 change huamn to human 

Spelling mistake has been fixed

Line 247 highly pathogenic avian influenza is a state only in birds (definition), HOAIVs can infect humans, zoonotic, but the HP status is not relevant in humans 

We have removed highly pathogenic and replaced this with H5N1

Line 316 Ifitm3 should be capital letters as per other instances, there is also a b instead of beta in the text, editorial checks 

We thank the reviewer for this and have fixed these mistakes.

conclusions section, influenza A/B or both refine 

We have specified the viruses in the conclusions
